# Associations of Children’s Close Reading Distance and Time Spent Indoors with Myopia, Based on Parental Questionnaire

**DOI:** 10.3390/children9050632

**Published:** 2022-04-28

**Authors:** Olavi Pärssinen, Essi Lassila, Markku Kauppinen

**Affiliations:** 1Department of Ophthalmology, Central Hospital of Central Finland, 40620 Jyvaskyla, Finland; 2Gerontology Research Centre and Faculty of Sport and Health Sciences, University of Jyväskylä, 40100 Jyvaskyla, Finland; markku.a.kauppinen@jyu.fi; 3Department of Ophthalmology, University of Helsinki and Helsinki University Hospital, 00029 Helsinki, Finland; essi.lassila@outlook.com

**Keywords:** parents opinion, children’s myopia, close reading distance, time spent indoors, outdoors, parents’ myopia, questionnaire study

## Abstract

Purpose: To study the association of parents’ reports about their children’s near work and outdoor habits with myopia in their children. Methods: Data from a questionnaire study conducted in 1983 among Finnish schoolchildren were reanalyzed. Vision screening had been performed for all the schoolchildren (n = 4961) in the 1st, 5th, and 8th grades (7-, 11-, and 15-year-olds) in an area of Central Finland. The questionnaire, including information about myopia, was returned by 4305 (86.7%) participants. Items concerned parents’ estimates of their child’s habitual reading distance, time spent indoors as compared with age peers, daily near work, outdoors time, and parents’ myopia. The associations of myopia with these factors were studied. Results: Myopia prevalence in those with a habitual close reading distance vs. others was 14.3% vs. 2.1%, 28.7% vs. 13.1% and 45.8% vs. 24.7% for the 7-, 11- and 15-year-olds (*p* < 0.001 in all age-groups). Myopia prevalence in children reported by their parents as spending more time indoors than age peers was 10.9% vs. 2.8% (*p* < 0.001), 25.0% vs. 14.7% (*p* = 0.004) and 41.9% vs. 25.7% (*p* < 0.001) in the three age groups. Myopia prevalence among those reported as spending both more time indoors and reading at a close distance vs. others was 44.2% vs. 11.9% (Fisher’s exact *t*-test, *p* < 0.001). In the multiple logistic regression models, parental myopia almost doubled the risk of myopia in the 11- and 15-year-olds. ORs (95% CI) for myopia adjusted for parental myopia and sex were for close reading distance 7.381 (4.054–13.440), 2.382 (1.666–3.406), 2.237 (1.498–3.057), (*p* < 0.001), and for more time spent indoors, 3.692 (1.714–7.954), *p* = 0.001, 1.861 (1.157–2.992), *p* = 0.010), 1.700 (1.105–2.615), *p* = 0.016, in the three age groups. Conclusion: Children, especially 7-year-olds, reported by their parents as having a close reading distance and spending a lot of time indoors were associated with a higher risk for myopia.

## 1. Introduction

The prevalence of near-sightedness (myopia) has been found to be increasing in many countries, with the highest increases being reported in many East and Southeast Asian countries, such as China, Singapore, Taiwan, Japan, South Korea and Hong Kong [1,2,3,4]. The rising global rate of myopia and its associated complications of cataract, glaucoma, retinal detachment and chorioretinal abnormalities is becoming a costly global health problem. There is no safe threshold level of myopia for any of its known ocular complications, and the higher the myopia, the higher the risk of complications associated with myopia [5,6,7]. Myopia is also the most common global cause of moderate and severe visual impairment and the second most common cause of blindness [2,5].

Many studies have confirmed that near work of various kinds [8], a short viewing distance [9,10] and parental myopia [11,12,13] are risk factors for myopia. Nowadays, children spend more and more time on mobile devices. High use of mobile devices has been found to increase the risk for myopia [14,15]. In a Chinese study of preschoolers, the use of mobile devices at the age of 1–3 years increased the risk of myopia at preschool age by more than 2.5-fold [16]. In contrast, time spent outdoors has been found to have a protective effect on myopia [10,17,18,19]. A recent study showed that time spent outdoors decreased the risk of myopia at different amounts of time spent in near work, although the decrease was lowest in the group spending the highest amount of time in near work [12].

This study analyzed the extent to which parents’ reports of their children’s behaviors was associated with myopia in their children. It is also discussed whether parents could contribute to lowering the risk of their children becoming myopic.

## 2. Material and Methods

Additional analyses were performed on old data originally obtained in a questionnaire study with schoolchildren in 1983 [20].

This study included all the schoolchildren (Finnish, Caucasians) (n = 4961) in the 1st, 5th, and 8th grades of basic school, henceforth 7-, 11- and 15-year-olds, living in an area of Central Finland in 1983 [20]. School nurses measured visual acuity at 5 m using an E-chart, without and, if worn, with spectacles [12]. After screening, the children were given a questionnaire to be completed together with their parents [12]. In 49 cases (1.1%), myopia status could not be confirmed and those cases were excluded. The number of questionnaires issued and returned is shown in Table 1.

The questionnaire asked parents (among other things) to state the child’s sex, whether either or both parents were myopic, to estimate whether their child spent more time indoors than age peers (more, same, less, can’t say), their child’s habitual reading distance (close, normal or long), the amount of time their child spent in reading and other near work outside school (sum of these = near work time) and in outdoor activities and sports excluding time spent at school (= outdoors time) [12]. The annual vision screening was done for all the children by experienced school healthcare professionals. Myopia in the children was defined based on information obtained from the vision screening by school healthcare professionals prior to issuing the questionnaire. The criteria for myopia were poor distant vision (≤0.7, Snellen notation) and good near vision without spectacles, and if spectacles were worn, whether they improved distant vision but not near vision [12]. Parental myopia was similarly defined in the questionnaire. Parental myopia was categorized into two groups: no myopic parents, and one or both parents myopic. One or both parents were myopic in 35.2% of cases [12].

### Statistical Analyses

The significance of differences between the categorical variables was tested by cross-tabulation and the differences between the discrete variables (e.g., myopia prevalence) by Chi-square test (Fisher’s exact test for 2 × 2 tables). A Student’s *t*-test was used to compare groups in terms of the amount of time spent indoors and outdoors. Predictors of myopia were studied using multiple logistic regression models [12]. Gender, parental myopia, time spent in indoors activities compared to age peers and reading distance were used as predictors in the models. General statistical analyses were performed using IBM SPSS statistics 26 (SPSS Inc., Chicago, IL, USA) and Stata version 12.0 (Stata Corp., College Stations, TX, USA). The level of statistical significance was set at *p* < 0.05 (two-sided).

## 3. Results

Table 2 shows the distribution of the answers to the two questions: (1) what is your child’s habitual reading distance; and (2) does your child spend more time indoors than his/her age peers?

### 3.1. Child Myopia according to Parents’ Estimates of Their Child’s Reading Distance

Reading distance was reported as close in 521 children and the prevalence of myopia in this group was 29.7%. Reading distance was reported as normal in 3685 children and the prevalence of myopia was 12.8%. Reading distance was reported as distant in 11 children, of whom one (8.3%) was myopic. The statistical difference in myopia prevalence was highly significant (Chi-square, *p* < 0.001) between the close and other reading distance groups but non-significant (Chi-square, *p* = 0.645) between the normal and distant reading distance groups. In the further analyses, those with a close reading distance were compared with those in the other two reading distance groups (normal and distant combined).

No significant differences were observed in daily time spent in near work or outdoors between those with a close or normal reading distance among the 7- and 11-year-olds. However, in the 15-year-olds with a close reading distance, their daily near work time was higher (2.50 (±1.17) vs. 2.22 (±1.10) h, *p* = 0.003), and their time spent outdoors lower (0.99 (±0.08) vs. 1.01 (±0.03), *p* = 0.003) than in normal distance group.

The prevalence of myopia in the close reading distance group was significantly higher across all the age and gender groups (Table 3). The greatest relative difference in the myopia prevalence between the age groups was found in the 7-year-olds. Myopia in close readers was about 7-fold more common this group than in the other age groups. In the 11- and 15-year-olds, a close reading distance increased the risk of myopia more in boys than girls.

### 3.2. Child Myopia according to Parents’ Estimates of Their Child’s Indoors Time

In the whole sample, the prevalence of myopia was significantly higher among those estimated to be spending more time indoors (26.5%) than those in the same (13.6%), less (12.4%) or can’t say (14.4%) groups (Chi-square, *p* < 0.001). The difference in myopia prevalence between the same, less and can’t say groups was non-significant (Chi-square, *p* = 0.537). Hence, further analyses were performed between two groups, recoded as (1) those estimated as spending more time indoors than peers and (2) the others.

Among those estimated as spending more time indoors, mean daily near work time was higher than among the others (2.81 (±1.27) vs. 2.24 (±1.01) h, *p* < 0.001) and mean daily outdoors time was lower (1.75 (±0.93) vs. 2.60 (±0.98) h, *p* < 0.001). Thus, the children reported by their parents as spending more time indoors than their age peers also spent more time doing near work and less time outdoors.

The children reported by their parents as spending more time indoors than their age peers were significantly more likely to be myopic (Table 4).

This association was found for all the age and gender groups, except the 11-year-old girls. Among the 7-year-olds, the prevalence of myopia among those spending a lot of time indoors was nearly four-fold as high compared to age peers spending less time indoors. Across all the age groups, myopia prevalence among those spending more time indoors was about double that of those in the other indoors time categories.

Among all the children spending more time indoors and reading at a close distance, the prevalence of myopia was 44.2% (23/52) compared to only 11.9% among the others (407/3240), (Fisher’s exact test, *p* < 0.001).

### 3.3. Adjusted Risk Factors for Myopia

Sex, parents’ myopia, close reading distance and time spent indoors compared to age peers were used as predictors in multiple logistic regression models for different age groups (Table 5).

Parental myopia almost doubled the risk of myopia in the 11- and 15-year-olds, but not among the 7-year-olds. Those with a close reading distance were at a higher risk of myopia than those spending more time indoors. A close reading distance increased the risk of myopia by about seven-fold in the 7-year-olds and about doubled the risk in the 11- and 15-year-olds. The risk for myopia among those spending more time indoors and reading at a close distance was greater in the 7-year-olds than in the older children. However, the risk for myopia of spending more time indoors and reading at a close distance differed little between the 11- and 15-year-olds. The myopia risk in the 11- and 15-year-old girls was about twice that of same-age boys.

## 4. Discussion

It must be noted that the habitual reading distance was based only on the parents’ subjective opinion and not on any measurements; if parents thought that the child is reading from a close distance, it can be suggested that they just deemed it to be too close. According to several studies, short near work distance (<20–30 cm), insufficient lighting and close-up viewing lasting more than 30–40 min without breaks, and low outdoor time are risk factors for myopia [21,22]. These issues deserve attention, especially in children.

This study showed that children reported by their parents as having a close reading distance and spending more time indoors than peers were at a four-fold risk for myopia. The study also showed that a close reading distance was associated with more time spent on close work and that a lot of time spent indoors was associated with less time spent on outdoor activities. Both more time spent in near work and less time spent outdoors are well known risk factors for myopia [8,23].

It is noteworthy that in this study close reading distance and spending more time indoors increased the risk of myopia most in the youngest children, aged 7 years. The younger the age of myopia unset, the faster is its progression and the greater the risk for high myopia (-6 D or more). The higher the myopia the greater the risk for vision treating complications, such as myopic macular degeneration (MMD), retinal detachment (RD), cataract, open angle glaucoma (OAG), and blindness [7,24].

In a Finnish study the prevalence of high myopia in adulthood was 65% if the first spectacles for myopia were received in the 3rd grade of school (aged 8.8–9.7 years), and only 7% if the spectacles were received two years later in the 5th grade [25]. The natural reading distance in children is closer the younger the age of the child. For example, in a follow-up study among myopic children, the mean reading distance of 11-year-olds was 19 cm and, 3 years later, 27 cm [9]. It is unclear why the present 7-year-olds who had a closer reading distance and spent more time indoors than their peers were at a higher risk for myopia than the 11- and 15-year-olds. Animal studies have shown that deprivation myopia caused faster axial elongation in younger than older animals [26]. Whatever the reason for the higher susceptibility to myopia of younger children, restricting prolonged near work at close distances, especially in young children, can be recommended.

In this study, more indoors time increased the risk for myopia. It is known that increasing outdoors time has a significant impact on preventing myopia. Time spent outdoors has been shown to reduce the incidence of myopia in children with different habitual amount of near work time [12], and to also prevent the myopic progression [10].

Sherwin et al. found that the pooled OR for myopia was 0.87 for an additional hour of time spent outdoors each day, indicating that increased outdoors time may be a simple strategy for reducing the risk for developing myopia and its progression in children and adolescents [27]. Wu et al. conducted a cluster-randomised controlled trial in which schoolchildren were encouraged to go outdoors for up to 11 h weekly and to take 10 min break after 30 min of near work [28]. At the one-year follow-up, the myopic shift was significantly less in the intervention group in both the non-myopic and myopic children [28]. In their 2-year follow-up study, Huang et al. found that a greater distance when doing near work, taking a break from near work every 30 min, and having more outdoor time were protective behaviors against myopia prevalence and progression over 6–24 months [29].

Myopia has increased worldwide, and it is now a major public health issue, especially in parts of East and Southeast Asia, including mainland China. In this region about 80% of students completing 12 years of school education are now myopic, and from 10% to 20% have high myopia [30]. The WHO meeting in 2015 expressed serious concern about the world-wide increase of myopia [31]. The meeting report predicted that in 2020 about one third of the worlds’ population would be myopic (2584 million) and that if this trend continues, close to 5 billion people would be myopic in 2050 [31]. Interventions to prevent myopia onset by increasing outdoor time have now been implemented at a system-wide scale in Chinese Taipei (Taiwan) and Singapore [30]. In recent years, mainland China’s President Xi Jinping has placed renewed emphasis on the prevention of myopia in China. In addition, China now seems to be aiming for major reform in schooling, including reducing academic pressure, particularly in the early school years, and ridding more time for outdoors play. In July 2021, China announced new regulation for private tutoring companies, an industry over USD 120 billion, obligating them to operate on a non-profit basis and limiting their operating hours. The aim is to reduce the near work load of schoolchildren and decrease the myopia risk [30].

Parental myopia is a well-known risk factor for myopia. In the meta-analysis of Zhang et al. the child's risk for myopia was about two to three times if one or both parents were myopic [13]. In the study of Tang et al. parental myopic status alone accounted for 11.8% of myopia variation in children, but the risk was 11.22-fold when both parents were highly myopic [32]. In their study more reading time increased, and more time outdoors decreased, the risk for myopia [32]. The attitudes and advice of parents play a major role in children’s behaviour and thus also in their health. Zhou et al. studied parents’ attitudes and behaviours towards children’s visual care and risk for myopia in school-aged children [33]. The results showed that the parents of the non-myopic children paid more attention to their children’s near work hours, stopped them using electronic devices in dim light, rectified their sitting posture when doing homework, took them to participate in outdoor activities, and ensured that they had sufficient sleep [33]. This indicates the importance, especially for myopic parents, of guiding their children in relation to the amount of time they spend on near work and outdoor activities.

Traditionally, parents have warned children about various behaviours that are widely regarded as harmful to health. Parents commonly admonish their children by telling them that reading in the dark or from too close a distance will ruin their eyesight. In the case of myopia, there may be an empirical basis for these warnings. The results of this study highlight the role of parents in reducing certain risk factors for myopia. Parents should be encouraged to ensure that their children do not engage in prolonged, uninterrupted reading or other near work at unnecessarily close distances. Parents should also provide children with opportunities for engaging as much as possible in outdoor activities.

## 5. Limitations

This is a cross-sectional study and therefore causal relationship could not be determined. The data analyzed in the study were drawn from a questionnaire study conducted about 40 years ago [12,20]. Thereafter, children’s habits of near work have been changed; watching television decreased and the use of smartphones and mobile devices increased. This change is not, of course, reflected in the present results. However, like reading, the use of smart phones and mobile devices has also been shown to be a risk factor for myopia [15,17].

A further limitation was the definition of myopia, which was based on information obtained by screening for distant vision at school, and questionary information about poor or good distant vision [12]. Myopia is the main cause of poor distant vision in schoolchildren [34,35], but hyperopia and astigmatism may potentially have confounded the results, as they often also cause difficulties in near vision. By comparing the results of this study with similar studies from Finnish schoolchildren and adults conducted using cycloplegia during the same period, it was possibly to calculate that the maximal limit of error in defining myopia in this study has been within 15% [12,36].

When the amounts of near work and outdoor time did not include the time spent in these activities at school, then the amounts of both these variables have been higher for each student. For this reason, the amounts of near work and outdoors time variables cannot be directly compared to other studies.

## 6. Conclusions

Children, especially 7-year-olds, reported by their parents as having a close reading distance and spending a lot of time indoors were associated with higher risk for myopia.

## Figures and Tables

**Table 1 children-09-00632-t001:** Study subjects.

Age, Years	Questionnaires Issued, N	Questionnaires (Including Data for Categorization for Myopia) Returned, N (%)		
		Boys, N (%)	Girls, N (%)	All, N (%)
7	1, 716	781 (50.0)	781 (50.0)	1562 (91.0)
11	1, 494	708 (51.7)	661 (48.3)	1369 (91.6)
15	1, 751	640 (46.6)	732 (53.4)	1372 (78.4)
All	4, 961	2129 (49.5)	2174 (50.5)	4303 (86.7)

**Table 2 children-09-00632-t002:** Distribution of parent’s estimates of their child’s reading distance and indoors time compared to age peers.

Reading Distance	N	%
Close	511	11.9
Normal	3687	85.7
Distant	12	0.3
Missing	95	2.2
Total	4303	100
**Indoors time compared to age peers**	**N**	**%**
More	308	7.2
The same	1939	45.0
Less	419	9.8
Can’t say	1584	36.8
Missing	43	9.8
Total	4303	100

**Table 3 children-09-00632-t003:** Prevalence of child myopia according to parents’ estimates of their child’s close reading distance.

Age, Years	Boys		Girl		All	
	Close Distance n/N (%)	Normal Distance n/N (%)	Close Distance n/N (%)	Normal Distance n/N (%)	Close Distance n/N (%)	Normal Distance n/N (%)
7	10/70 (14.3)	14/682 (2.1)	11/77 (14.3)	15/681 (2.2)	21/147 (14.3)	29/1363 (2.1)
Fisher’s exact test, *p*	**<0.001**		**<0.001**		**<0.001**	
11	15/67 (22.4)	59/629 (9.4)	45/142 (31.7)	90/507 (17.8)	60/209 (28.7)	149/1136 (13.1)
Fisher’s exact test, *p*	**0.003**		**<0.001**		**<0.001**	
15	13/34 (38.2)	97/584 (16.6)	58/121 (47.9)	196/602 (32.6)	71/155 (45.8)	293/1186 (24.7)
Fisher’s exact test, *p*	**0.003**		**0.001**		**<0.001**	
7, 11, 15	38/171 (22.0)	170/1895 (9.0)	114/340 (33.5)	301/1790 (16.8)	152/511 (29.7)	471/3685 (12.8)
Fisher’s exact test, *p*	**<0.001**		**<0.001**		**<0.001**	

n/N = number of myopes/all subjects, (%) = prevalence of myopia. Significant *p*-values bolded.

**Table 4 children-09-00632-t004:** Prevalence of myopia according to parents’ estimates of whether their child spends more time indoors than age peers.

Age Years	Time Spent Indoors as Compared with Age Peers					
	Boys		Girls		All	
	More	Same or Less	More	Same or Less	More	Same or Less
	n/N (%)	n/N (%)	n/N (%)	n/N (%)	n/N (%)	n/N (%)
7	5/46 (10.9)	20/727 (2.8)	5/46 (10.9)	21/727 (2.9)	10/92 (10.9)	41/1454 (2.8)
Fisher’s exact test, *p*	**0.013**		**0.015**		**0.001**	
11	12/59 (20.3)	62/643 (9.6)	16/53 (30.2)	121/606 (20.0)	28/112 (25.0)	183/1249 (14.7)
Fisher’s exact test, *p*	**0.014**		0.061		**0.004**	
15	12/35 (34.3)	102/596 (17.1)	32/70 (45.7)	219/653 (33.5)	44/105 (41.9)	321/1249 (25.7)
Fisher’s exact test, *p*	**0.014**		**0.030**		**<0.001**	
7, 11, 15	29/140 (20.7)	184/1966 (9.4)	53/169 (31.4)	361/1986 (18.2)	82/309 (26.5)	545/3952 (13.8)
Fisher’s exact test, *p*	**<0.001**		**<0.001**		**<0.001**	

n/N = number of myopes/all subjects, (%) = prevalence of myopia. Significant *p*-values bolded.

**Table 5 children-09-00632-t005:** Multiple logistics regression models (OR, 95% confidence intervals) for myopia in different age groups adjusted by three factors.

Predictors	OR	95% CI	*p*
Age 7 (Model 1)			
Parental myopia (ref. no myopia) one or both parents myopic	0.826	0.453–1.505	0.532
Reading at a close distance (ref. normal or far distance)	**7.381**	4.054–13.440	**<0.001**
More time indoors than age peers (ref. same or less)	**3.692**	1.714–7.954	**0.001**
Girls (ref. boy)	1.036	0.579–1.853	0.906
Age 11 (Model 2)			
Parental myopia (ref. no myopia) one or both parents myopic	**1.884**	1.387–2.558	**<0.001**
Reading at a close distance (ref. normal or far distance)	**2.382**	1.666–3.406	**<0.001**
More time indoors than age peers (ref. same or less)	**1.861**	1.157–2.992	**0.010**
Girls (ref. boy)	**1.995**	1.454–2.737	**<0.001**
Age 15 (Model 3)			
Parental myopia (ref. no myopia) one or both parents myopic	**1.890**	1.437–2.486	**<0.001**
Reading at a close distance (ref. normal or far distance)	**2.237**	1.498–3.057	**<0.001**
More time indoors than age peers (ref. same or less)	**1.700**	1.105–2.615	**0.016**
Girls (ref. boy)	**2.237**	1.715–2.917	**<0.001**

Significant OR’s values are bolded.

## Data Availability

The data presented in this study are available on request from the corresponding author. The data are not publicly available due to privacy.

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
