# Peer review of "Associations of Children’s Close Reading Distance and Time Spent Indoors with Myopia, Based on Parental Questionnaire"

_children, 2022, doi:10.3390/children9050632_

Round 1
Reviewer 1 Report
First and foremost, some mention of adhering to the ethical principles of the Declaration of Helsinki must be made, even if the data are 40 years old and they have been de-identified. I would mention that.
This study is from a self-reported questionnaire which is one of the weaker forms of patient-oriented research. It's not that we cannot learn anything, but we must be careful with our interpretation.
For example, the report from parents that myopic children hold their reading material closer is not surprising. There is a chicken and egg question. Do myopic children hold their reading material closer because they are myopic and it is clearer up close, or does holding the material close cause myopia. I think the evidence is fairly clear that there is not a causal relationship between close work and myopia. We must be careful to not infer there that there is a causal relationship. In fact, the entire near work section could be deleted and it might make this paper better.
What has been clearly shown in the last 40 years is the relationship between less time spent outdoors and the incidence of myopia. Of this there is no doubt. In fact, if these data would have been published and discussed 40 years ago, it would have pointed the way for investigators. These results are interesting and any evidence of parents being able to guide the behavior of their children would be interesting to encourage children to spend more time outdoors within reason and when safe.
In its present form I would not publish these results. Publishing results that suggest that near work is a cause of myopia, based on questionnaire data, that is the opinion of people, would not be in the best interests of science.
Author Response
I totally disagree with the referee that there is no connection between near work and myopia.
I've been doing myopia research for over 40 years and at the same time been following research in this
field and attended regularly to international myopia congresses. For more than 100 years, epidemiological
studies have shown that connection and, at the same time, the myopia preventive influence of time spent
outdoors. In my randomized controlled clinical trial I was perhaps the first to demonstrate this connection
on myopic progression. My recent study showed that myopia in 7- and 11-year-olds was rare is daily near
work at home did not exceed one hour. Animal experiments also contribute to this, different methods to
preventing distant vision classically cause myopia in animals. I recommend some publications to the
transformers to be read.
Contrary to what the referee writes, it is very important that parents consider the risks of lengthy of near
work and short time in outdoor activities. Adequate outdoor activities have been found to have many of
the benefits associated with general health. These matters cannot be left to the school and the
administration alone, but parents have an essential health significance in this matter.
There are a large number studies about this, here are just a few examples.
Cohn H. Untersuchungen der Augen von 10060 Schoolkindern nebst Vorschlägen 3 fur Verbesserung der
Augen nachtheiligen Schul-Einrichtungen, Leipzig 1867. Cited by DS Rehm. The myopia myth. IMPA,
Ligonier, The United States 1981:57.
Tscherning M. Studien uber die Aetiologie der Myopie. Albrcht Von Graefe’s Arvch Ophthalmol
1988:29:201-272.
Pärssinen, O.; Lyyra, A.L. Myopia and myopic progression among schoolchildren: a three-year follow-up
study. Invest Ophthalmol Vis Sci. 1993; 34, 2794-2802.
Pärssinen, O.; Kauppinen, M. Associations of near work time, watching TV, outdoors time, and parents’
myopia with myopia among school children based on 38-year-old historical data. Acta Ophthalmol 2022
https://doi.org/10.1111/aos.14980.
Zhi ZN, Yang TZ, Xiong SB, Jiang LQ, Pan MZ, Qu J & Zhou XT (2010): Susceptibility of guinea pig eyes to form
deprivation myopia and its age-related recovery. Zhonghua Yan Ke Za Zhi 46: 641-645.
Hansen, M.H.; Laigaard, P.P.; Olsen, A.M.; Skovgaard, A.M.; Larsen, M.; Kessel, L.; Munchet, I.C. Low
physical activity and higher use of screen devices are associated with myopia at the age of 16-17 years in
the CCC2000 Eye Study. Acta Ophthalmol. 2020, 98, 315-321.

Reviewer 2 Report
Vision testing and defining myopia are questionable. Not using cycloplegia, may overestimated myopia. Testing visual acuity in children may be influenced by their compliance.
Another question raised: what was the normal/ close/far child’s habitual reading distance?
Author Response
The annual vision screening was done for all the children by experienced school nurse. It has been added to the text.
It must be noted that the habitual reading distance was based only on the parents’ subjective opinion and not on any measurements. If parents thought that child is reading from close distance, it can be suggested that they deemed it to be too closet. According to several studies, short near-work distance (< 20-30cm), insufficient lighting and close-up viewing lasting more than 30-40 minutes without breaks are risk factors for myopia [21,22]. These issues deserve attention, especially in children. The first sentence in the abstract has been added.
The reliability of the questionnaire answers on distant vision of children was controlled for by comparing these with the results of the vision test administered by the school nurses to a random sample of children (n = 354). The sensitivity of the questionnaire to poor distant vision (≤ 0.7) in this comparison was 86% and specificity 84%.
Round 2
Reviewer 1 Report
This a review of the first revision of this manuscript.
The conclusion of this paper is that, "Parents’ opinion of child’s close reading distance and spending much time indoors significantly predicted increased risk of myopia in their children, most in the 7-year-olds. The results suggest that parents can have a significant role in prevention of myopia, by encouraging children to avoid too short near viewing distance and ensuring them for adequate time for outdoor activities."
It matters little what parents believe to be the cause of myopia. What matters is the science. The science shows that genetics and exposure to bright light/sunlight help to prevent myopia. Near work has been shown to be of little consequence, or at best a confounding factor, meaning that children who read more indoors tend to get less outdoor exposure.
This manuscript sends the wrong message. My suggestions to drop or minimized the near work conclusions and emphasize the outdoor light message were ignored.
Author Response
Dear reviewer
The main criticism of the reviewer 1 was directed at the fact that the manuscript describes near-work as a risk factor for myopia.
I like to give my rebuttal to this statement.
As a justification, I would like at first to introduce myself (Olavi Pärssinen)
I've been doing myopia research for about 40 years and published quite a lot of articles on myopia. Currently my study group is collecting 40-year follow-up data of subjects whose myopia onset was at 3rd and 5th grades of school in 1983-1984 and whose changes of myopia has been followed-up about 25 years.
As an editorial board member of ACTA Ophthalmologica I have followed the current myopia research for years. In am also reviewing myopia studies for many different ophthalmologic journals.
When the Joint World Health Organization-Brien Holden Vision Institute Global Scientific Meeting on Myopia 2015 took place in Sidney, I was one of the 20 myopia researchers invited to the meeting. The conclusion of our meeting was published in 2016: “WHO report. The impact of myopia and high myopia: report of the Joint World Health Organization–Brien Holden Vision Institute Global Scientific Meeting on Myopia, University of New South Wales, Sydney, Australia, 16–18 March 2015.”
Since then I have been a member of IMI. The IMI are a global group of experts who have come together to discuss, debate and make available the latest evidence-based recommendations in classifications, patient management, and research, in the form of the IMI white papers.
I have been the member of genetic study group, Consortium for Refractive Error and Myopia (CREAM) from its start. The research collaboration comprising more than 50 international research groups. During the years CREAM has published several studies about genetics of refraction and myopia, a part these with collaboration with 23andMe and UK Biobank.
As examples of these studies, here some with short conclusions:
Tedja , Wojciechowski, Hysi et al. Genome-wide association meta-analysis highlights light-induced signaling as a driver for refractive error. Nat Genet. 2018 Jun; 50(6): 834–848.): The results support the notion that refractive errors are caused by a light-dependent retina-to-sclera signaling cascade. However, only about 20 % of the variation of refraction and about 8% of myopia could be explained by genetics.
Tideman, Pärssinen, Haarman et al. Evaluation of Shared Genetic Susceptibility to High and Low Myopia and Hyperopia. JAMA Ophthalmol. 2021 Jun 1;139(6):601-609. doi: 10.1001/jamaophthalmol.2021.0497. Genetic risk variants were shared across HM, LM, and hyperopia and across European and Asian samples and individuals with HM inherited a higher number of variants from among the same set of myopia-predisposing alleles and not different risk alleles compared with individuals with LM.
Qiao Fan, Xiaobo Guo, J. Willem L. Tideman, Katie M. Williams, et al. Childhood gene-environment interactions and age-dependent effects of genetic variants associated with refractive error and myopia: The CREAM Consortium. Scientific Reports 2016, | 6:25853 | DOI: 10.1038/srep25853. There was no indication that variant or GRS effects altered depending on time outdoors, however 5 variants showed nominal evidence of interactions with near work.
It is true that parental myopia clearly increases the risk of myopia in their children, two myopic parent more than one. On the other hand, the myopic parent are more educated, which is a significant risk factor of myopia. Thus, parental myopia does not directly mean heredity and is not good indicator for genetics of myopia.
There is plenty of studies showing the connection between different types of near works and incidence of myopia. However, studies showing connection between near work and progression of myopia are fewer. The younger is child the greater is risk of child to be myopic when doing much near work. The intermediate viewing distance, for example watching TV from a few meters distance, neither increase of decrease the risk of myopia.
There is no doubt that time spent outdoors prevent incidence of myopia and also progression of it. According to my latest report the protective influence of time spent outdoors was seen at different levels of near work time, but relatively less at the highest level of near work. work.(https://onlinelibrary.wiley.com/doi/full/10.1111/aos.14980)
However, no evidence, as far as I know, is supporting the reviewers view that the lack of outdoor activities would be the cause of myopia, but there is no doubt that it is a preventive factor. According to similar logic, for example infectious diseases or allergy would be caused by lack of medicines.
I cannot misrepresent the results or neglect the essential results from our manuscript.
In order to better take into account the influence parental myopia on myopia in their children, we have added the following sentence to the abstract: In the multiple logistic regression models, parental myopia almost doubled the risk of myopia in the 11- and 15-year-olds.
We hope that manuscript in its current content would help to draw parents' attention to children's use of time in order to prevent myopia and would thus be suitable to be published in Children.
Reviewer 2 Report
no comment
Author Response
Thanks for the review.